# Health care during electricity failure: The hidden costs

Abigail Mechtenberg[1,2¤]*, Brady McLaughlin[1,2], Michael DiGaetano[1,2],
Abigail Awodele[1,2], Leslie Omeeboh[1,2], Emmanuel Etwalu[3], Lydia Nanjula[4],
Moses Musaazi[3†], Mark Shrime[5]

**1** Center for Sustainable Energy, University of Notre Dame, South Bend, Indiana, United States of America,
**2** Department of Physics and Preprofessional Studies, College of Science, University of Notre Dame, South
Bend, Indiana, United States of America, **3** College of Engineering, Design, Art, and Technology (CEDAT),
Makerere University, Kampala, Uganda, **4** Mulago Hospital, Kampala, Uganda, **5** Harvard Medical School,
Harvard University, Boston, MA, United States of America

† Deceased.
¤Current address: Physics Department, University of Notre Dame, Notre Dame, Indiana, United States of
America
* amechten@nd.edu

org/10.1371/journal.pone.0235760

Molise, ITALY

**Data Availability Statement:** All relevant data are
within the paper and its Supporting information
files.

**Funding:** University of Notre Dame's Center for
Sustainable Energy has given ESDD Lab Seed

## Abstract

### Background

Surgery risks increase when electricity is accessible but unreliable. During unreliable electricity events and without data on increased risk to patients, medical professionals base their decisions on anecdotal experience. Decisions should be made based on a cost-benefit analysis, but no methodology exists to quantify these risks, the associated hidden costs, nor risk charts to compare alternatives.

### Methods

Two methodologies were created to quantify these hidden costs. In the first methodology through research literature and/or measurements, the authors obtained and analyzed a year's worth of hour-by-hour energy failures for four energy healthcare system (EHS) types in four regions (SolarPV in Iraq, Hydroelectric in Ghana, SolarPV+Wind in Bangladesh, and Grid+Diesel in Uganda). In the second methodology, additional patient risks were calculated according to time and duration of electricity failure and medical procedure impact type. Combining these methodologies, the cost from the Value of Statistical Lives lost divided by Energy shortage ($/kWh) is calculated for EHS type and region specifically. The authors define hidden costs due to electricity failure as VSL/E ($/kWh) and compare this to traditional electricity costs (always defined in $/kWh units), including Levelized Cost of Electricity (LCOE also in $/kWh). This is quantified into a fundamentally new energy healthcare system risk chart (EHS-Risk Chart) based on severity of event (probability of deaths) and likelihood of event (probability of electricity failure).

### Results

VSL/E costs were found to be 10 to 10,000 times traditional electricity costs (electric utility or LCOE based). The single power source EHS types have higher risks than hybridized EHS

Funding to do research in Sustainable Energy and International Development broadly. University of Notre Dame paid Dr. Mechtenberg summary salary as SEED Funding for her new ESDD research lab. Undergraduate research students received either Slatt Fellows research funds or ESDD work study research funds: Brady McLaughlin, Michael DiGaetano, Abigail Awodele, and Leslie Omeeboh. The funders had no role in study design, data collection and analysis, decision to publish, or preparation of the manuscript.

**Competing interests:** No authors funded to do this specific research project due to the fact that funding agencies we approached do not fund this type of research connecting electricity and health care directly.

types (especially as power loads increase over time), but all EHS types have additional risks to patients due to electricity failure (between 3 to 105 deaths per 1,000 patients).

## Conclusions

These electricity failure risks and hidden healthcare costs can now be calculated and charted to make medical decisions based on a risk chart instead of anecdotal experience. This risk chart connects public health and electricity failure using this adaptable, scalable, and verifiable model.

## 1. Introduction

Over 1 billion people in the world lack access to electricity [1, 2]. In low-income and middle-income countries (LMIC's), this global epidemic is especially poignant in health care delivery, or lack thereof [3–15]. As an integral part of society, electricity serves as a vital component of health care policies [14, 16, 17]. Without reliable access to electricity, many necessary components of a hospital, such as lights, anesthesia machines, and imaging equipment, become ineffective and unusable, especially in times of urgent medical needs [18, 19]. The Lancet Commission on Global Surgery, the World Health Organization, and many others states that access to reliable electricity serves as a potent factor in global surgery [4, 6, 20–22]. The impact of electricity failures threaten the ability for the development and delivery of surgical, anaesthesia, and overall healthcare in LMICs, as seen in Fig 1. Its effects can range from postponing surgery, postponing accurate diagnoses for a needed surgery, permanent disabilities, and even to fatalities during surgery, due to failure of various medical equipment [1, 2, 6, 23–27]. These effects are additional risks to patients that accompany medical procedures, especially surgeries [28], when electricity fails. This puts medical professionals in a situation of possible ethical dilemmas such as either (1) starting a necessary surgery to save a life knowing there is risk of electricity failure, or (2) delaying surgery due to electricity failure risk. At times, there can be two or more ethical scenarios to consider. [27] Without any data or quantifiable models on which to base medical decisions, it is difficult to evaluate alternatives or solutions. Although there is a lack of data, there exists a gamut of anecdotal experiences [6, 23, 29, 30] from experts on such ethical dilemmas [19].

There also exists data on electricity outages for various energy healthcare systems (EHS) types and countries [16, 31–34]. This data and other publications focus on the impact of unreliable electricity as it relates to a health care facility or health care service [35, 36], but typically not on medical procedures. Even with the implementation of these optimal solutions and these conceptual frameworks, as presented in the research literature, electricity failures endure without action plans and leave medical professionals with choices that turn into ethical dilemmas. More importantly, the research literature lacks clearly articulated and verifiable potential methodologies to evaluate the ethical dilemmas for various EHS when electricity fails for specific medical procedures. Thus, the authors have created a model that generates risk charts, based on frequency of electricity failures and consequences to health care delivery. This methodology links energy system configurations and health care consequences in a manner that is adaptable, scalable, and verifiable.

## 2. Methods to bridge health care and energy system

This section describes combining two methodologies: modeling electricity failure and modeling additional patient risk. The first methodology deals with the time of electricity failure at

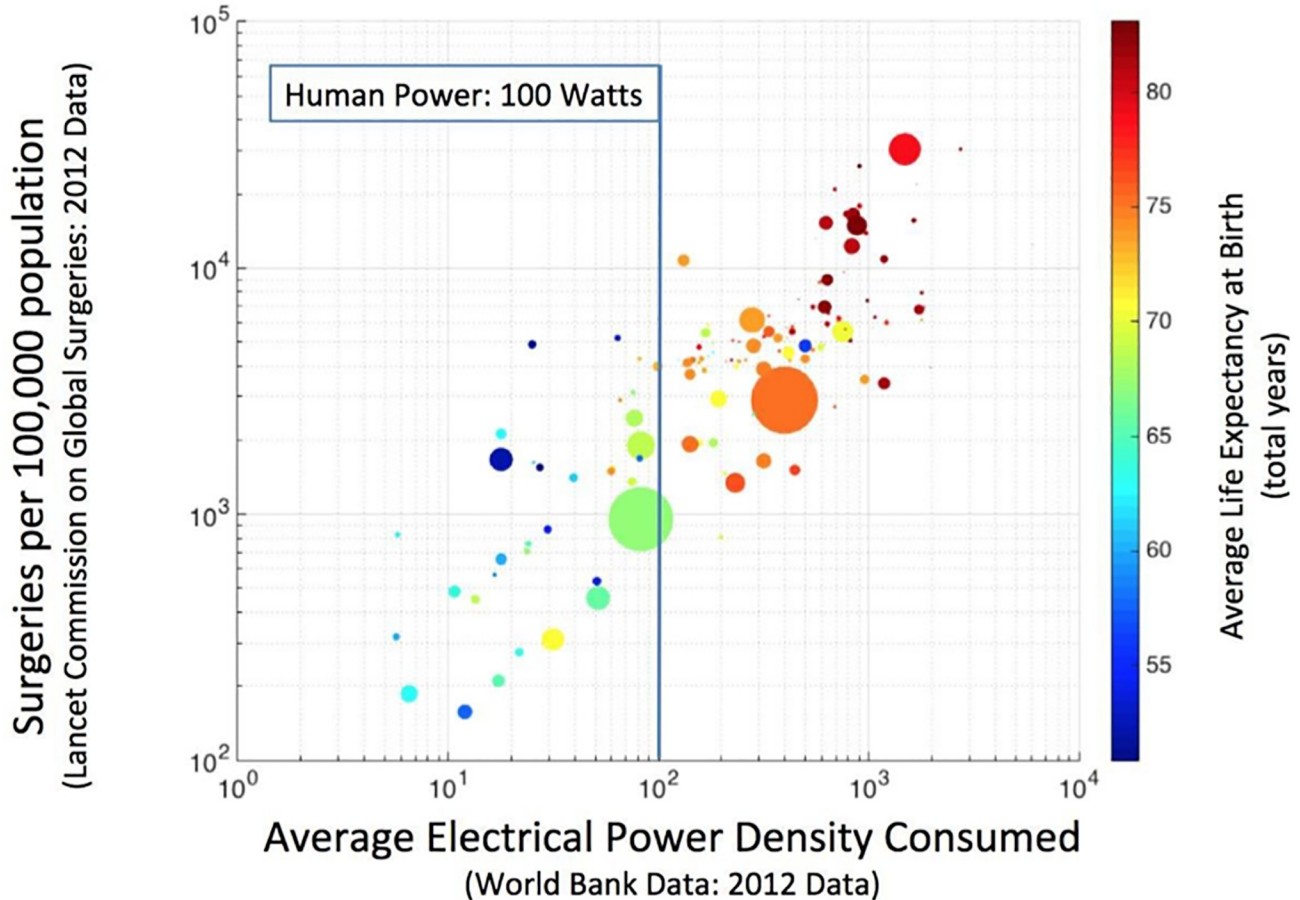

**Fig 1. Global surgeries per 100,000 population is not equitably distributed around the world, dramatically interfering with average life expectancy at birth (colored data) plotted by country (size of circle is a country's population).** As global surgeries increase, average electrical power density consumed (2012 data on x-axis in units of Watts/Capita) must increase to ensure accessibility and reliability to health care facilities adding surgical wards.

specific health care facilities. The second methodology quantifies additional patient risk for medical procedures, vis-a-vis time and duration of electricity failures, for the purpose of modeling risk charts. Using research literature, the authors obtained hour-by-hour energy failure events for three health care facilities, from Homer Energy optimal models, in Iraq [16], Bangladesh [31], Ghana [32, 33], and a health care facility in Uganda [34, 37], with failure rate data. Three of the papers, cited above, published solutions for EHS 1,2,3. These solutions are presented as optimal, or necessary, given the political climate of the location. Furthermore, they presented and accepted a capacity shortage of 10-50%. These solutions were replicated and analyzed to delineate why such huge margins in capacity shortage were accepted, given the consequences to patients. The fourth system, EHS 4, in Uganda is based on actual measurements of voltage and current over half a year [31]. The data from Uganda was used to create a contrast to modeled data. This is to say that it adds to the flexibility and applicability of the paper as it is not limited to modeled data. In this way, all the data were used to quantify and better understand the factors that contribute to accepting capacity shortage in an EHS. Sometimes, a 50% decrease in cost is argued as acceptable for a 20% capacity shortage [16, 34]. For

instance, an increase in power consumed, caused by accepting a donated medical electrical device [38, 39], can quickly increase an energy healthcare system's capacity shortage to 20%. This shortage is accepted as long as it decreases costs. Thereby, compounding consequences and risks to patients.

## 2.1 Energy healthcare system (EHS) failure methodology

Many grid and decentralized electrical energy systems in LMICs struggle with electricity failures for EHS, including availability of backup generators [40]. By compiling obtained energy data with state-of-the-art engineering modeling from research literature, the frequency and duration of electrical failures can be quantified and plotted across the year.

Fig 2 shows data retrieved from modeling a solar-powered health care facility with only an 18% energy capacity shortage. This data was published as optimal because of rational trade-offs and reasons. Furthermore, it suggests that the timing of the shortage will dramatically affect health care, in terms of single long duration power failures versus multiple small duration power failures. This is especially true in medical decision-making: If a surgeon knows the electricity will fail all day, then the medical decisions are different than if the electricity will fail in one or two hours and last only for one hour randomly. The figure, deceptively, suggests that early mornings during summer days may be more favorable to do medical surgeries. However, if powerload is increased at those times, the capacity shortage issue persists. Thus, it is still randomly unreliable.

Data compiled on capacity shortages are typically presented as a number, not as a yearly plot (Hours during the Day versus Days of the Year) or using known terminology associated with risk. This means that a 20% capacity shortage is presented as 80% electrical power availability, diverting analysis and decisions to cost savings, not the consequences of the 20% shortage. The medical consequences of these 20% electricity failures in a health facility will depend on the type of medical procedure, the time, duration and frequency of failure events, and time of day of events. Beyond Fig 2, the three other EHS types are discussed in detail in S1 File and shown in Fig 4. In the next subsection, the authors created a methodology to quantify the impact of failures by grouping the additional risks to patients based on time of electricity failure, duration of failure, and type of medical impact (no, low, medium, high impact).

## 2.2 Energy healthcare system (EHS) risk methodology

Patients, in medical procedures that depend on electricity, take on additional risks during unreliable electricity failure events. There is no data available for these additional risks. Research surveys suggest that the risks are considerable [9, 10]. Journalists have collected stories from doctors on patient deaths due to increased electricity failures at a regional hospital [41]. Medical experts admit basing decisions on anecdotal experience using qualitative scenarios and logic [19]. Consequently, the authors have created a quantifiable model of these additional patient risks, based on the type of medical procedure and the time and duration of electricity failure. The purpose of creating the model is to provide a tool that supports medical professionals in eliminating ethical dilemmas by modeling various scenarios and probabilities of impacts.

Consider being a doctor in a LMIC. You are faced with a decision: (1) turn off a diesel generator to cool it down, but in so doing accept the significant probability that 10 patients die now or (2) leave the generator on because the grid might come back on soon. However you risk the generator failing and the significant probability that patients also die in the future, due to compounded electricity failure due to the time it takes to replace a failed generator. What is the best decision? It is a troubling decision to make. At first, the medical professional chooses

# Energy Healthcare System Example

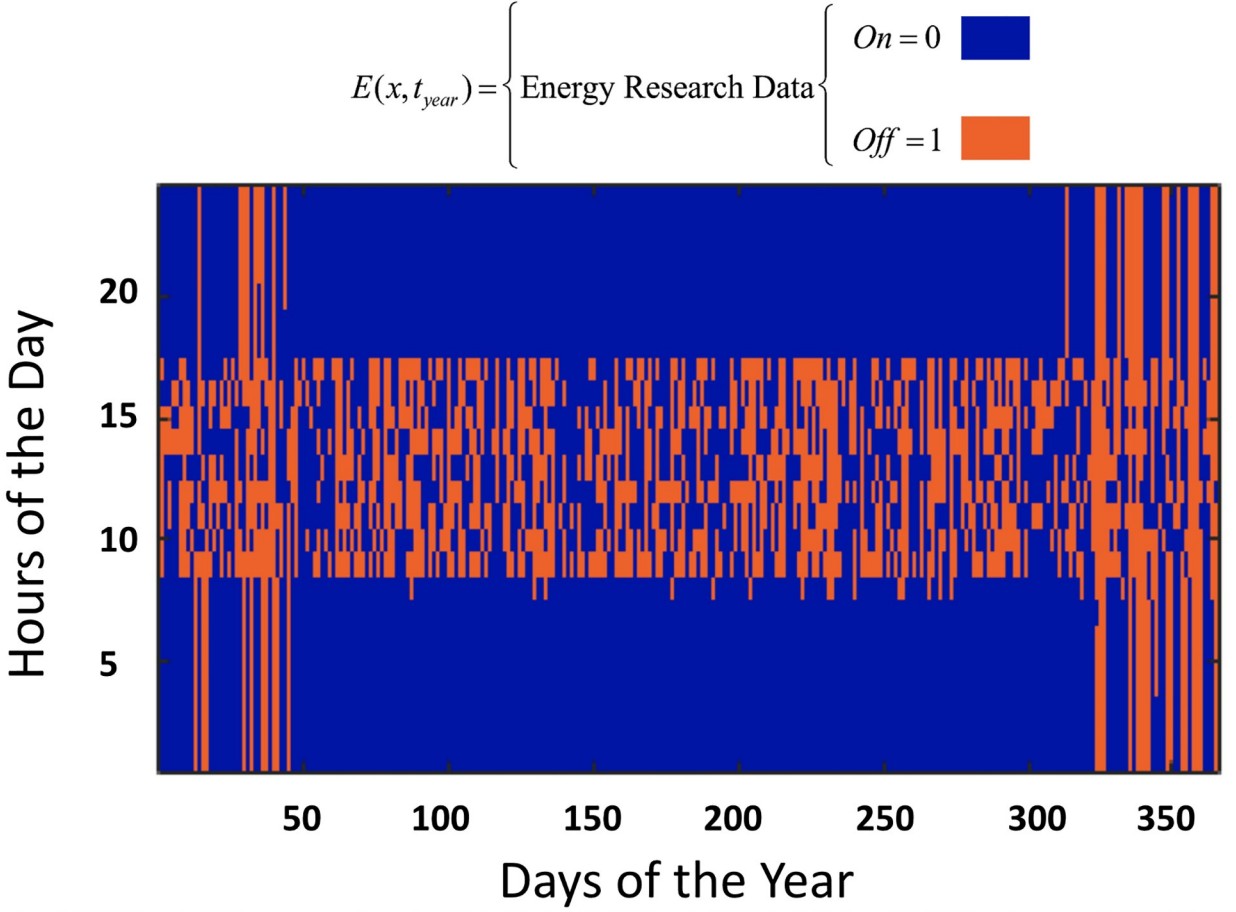

**Fig 2. lectricity failure events (red) for Iraq Rural health care facility published in Solar Energy, 2010 [16].** Hours of the day on the y-axis and days of the year on the x-axis with color showing state of electricity system: on (blue) or off (red).

option 2 and the diesel generator failed and it took six months to get an new diesel generator. Based on this experience, the medical professional chooses option 1 and the 10 patients died due to needing oxygen concentrators to breathe successfully (something that has happened in the United States in Puerto Rico [42] after the hurricane, and in California due to electricity outages [43]). Although other options may exist, they cannot be considered until risk modeling occurs enabling an understanding of additional patient risk due to electricity failure frequency and duration.

Four categories were created to quantify additional risk to patients associated with medical procedures: no impact, low impact, medium impact and high impact. The first category consists of all medical procedures which have no impact due to electricity failure. This is represented by a flat line (no increase or decrease in additional patient risk). The three categories of curves in Fig 3 represent the consequences of starting a medical procedure, requiring electricity, after which an electrical failure occurs, and the patient experiences additional risks quantified as high impact (in red), medium impact (in green), or low impact (in blue). Within each

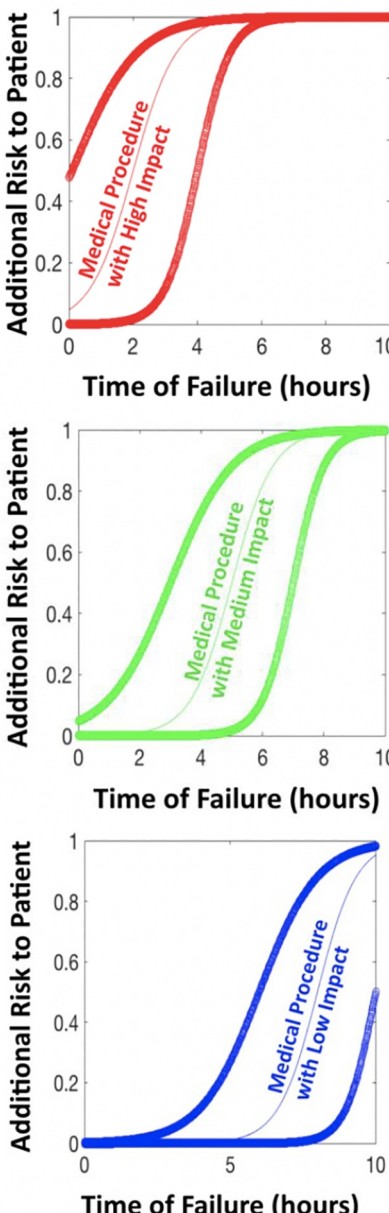

**Fig 3. The y-axis shows additional risk to patient as a result of starting a medical procedure which required an electrically powered medical device.** The duration of the electricity failure increases the patient risk, which is on the x-axis. Medical procedures can be placed into four categories of impact regions (high impact—top, medium impact—middle, and low impact—bottom) or no impact (not shown, but flat line on zero). These curves can also be easily adjusted with two parameters as data becomes available.

grouping, the central curve serves as the mean value, whereas the curves on either side of the central curve serve as the confidence intervals for each value. Future research will be able to identify specific medical procedures in the high, medium, low, or no impact group and how to scale the two parameters defining these curves, in order to easily fit data to this scalable model (c and k parameters described in S1 File). In reverse logic, medical professionals can alter these two parameters, and percentages of medical procedures done at the hospital, given an EHS-Type to calculate the total risk. If the data ever becomes available, then changes in the slope and height of curve, as well as the uncertainty range can decrease and become more precise and clearly verifiable. The risk function can also change completely, but the overall methodology remains the same. Beyond the logistical functions implemented here, the authors considered other two parameter probability functions as discussed in S1 File. Consequently, the overall adaptability of this methodology suggests that other additional patient risk functions can easily be compared and verified in future research, but the need for combining these two methodologies is vital and currently non-existent.

### 2.3 Combining two methodologies into one

The quantification of consequences to patients in a healthcare facility is calculated over a year and grouped into five risk severity levels: negligible, minor, moderate, significant, and catastrophic. The quantification of electricity failure events is calculated over a year and grouped into five likelihood levels: improbable, remote, occasional, probable, and frequent. Based on the number of days in a potential risky event over an entire year, grouped by severity and likelihood level pair, the hidden medical cost due to statistical lives lost is divided by the amount of electrical shortage needed to prevent this risk. However, the statistical lives lost depend on inputs from medical professionals in terms of the percentage of medical procedures in no impact, low impact, medium impact, and high impact categories, as well as the total number of patients. From Section 2.2, initially, additional statistical risk is calculated in units of deaths per 1000 patients versus Section 2.1 which calculates capacity shortage in energy units of kWh based on EHS data and/or state-of-the-art engineering models. Using the Value of a Statistical Life (VSL in $) [44, 45] and the electrical capacity shortage (E in kWh), the hidden energy cost based on lives lost is calculated and defined as VSL/E (in units of $/kWh) and compared to the traditional Levelized Cost of Electricity(LCOE) (in units of $/kWh) [7].

## 3. Hidden energy costs in healthcare results

There are two key results in this paper to show the breadth of this methodology in its adaptability, scalability, and verifiability. Section 3.1 presents the electricity failure for the unique EHS types and contexts, juxtaposed with corresponding additional patient risk: Iraq [16], Ghana [33], Bangladesh [16, 32], and Uganda [34] which will be defined as EHS-Type 1, EHS-Type 2, EHS-Type 3, and EHS-Type 4, respectively. The goal is to consider various types of electricity failures found in different energy healthcare system (EHS) types. Section 3.2 focuses on the VSL where the health care facility is located, the total electricity shortage based on energy system type, and on the new VSL/E hidden electricity costs. These VSL/E costs due to the loss of life and due to electricity failure are then compared to traditional cost of electricity, LCOE (both in $/kWh units).

### 3.1 Electricity failures and patient risk

There seems to be a tendency to present information in a manner that clouds transparency and minimizes understanding the consequences. For instance, EHS-Type 1, from Fig 4, represents the results of a solar panel powered energy system, which shows an 18% energy capacity

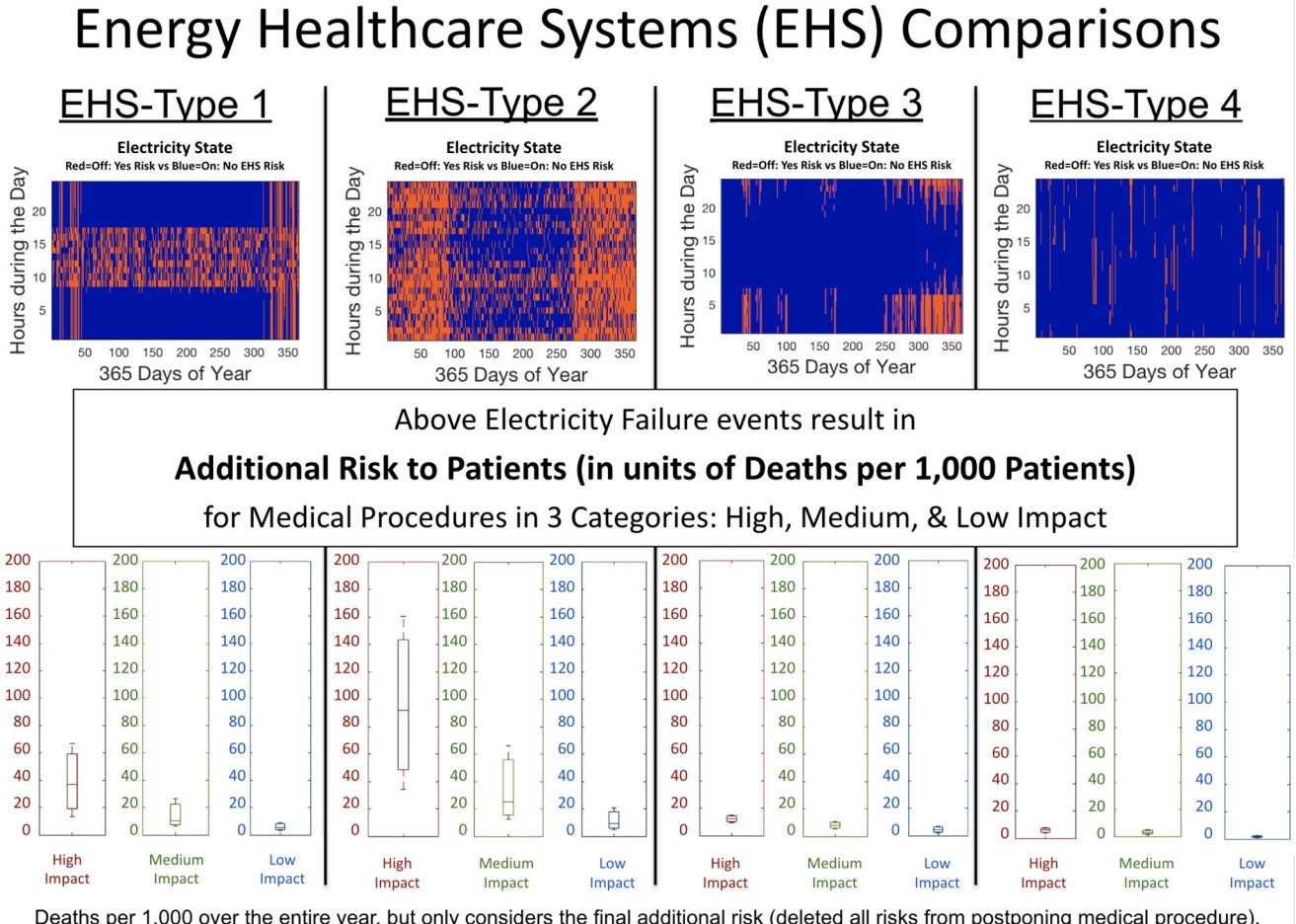

**Fig 4. Top figures show four health care electricity failure events for a year with electricity ON in blue (0) and electricity OFF in red (1).** The bottom figures show corresponding calculated deaths per 1,000 patients for the entire year for high impact, medium impact, and low impact medical procedures for each system depending on time and duration of electricity failure. Even with grid and backup diesel generator system at a regional hospital, with only 4% energy and time capacity shortages, there were additional risks to patients.

shortage. This presentation of capacity shortage makes it seem as only occurring at night or only 18% of the time. However, the shortage is mainly during the day and with 25% time capacity shortage. There are systems, as seen in the S1 File, that start with 0% capacity shortage and quickly rise to 20% energy capacity shortage due to small increases in power loads. Similarly, in EHS-Type 2, there is 11% energy capacity shortage. Although it sounds minimal, this amount of energy needed accounted for a capacity shortage 49% of the time (almost half the year). And this solution was presented as a political solution where the government provides electricity 50% of the time from hydroelectricity and the health care facility must acquire other ways to produce electricity for the rest of the time. In other words, both time and energy shortages matter. The difference between these two values underscores the need for medical and engineering professionals to request that all engineering electricity system designers present the time as well as the energy shortage in numbers and yearly maps and potentially work together to create additional risks to patients based on these designs and decisions as shown in Fig 4 below.

**Table 1. Qualitative pattern justification in choosing these four EHS types.**

| | Electricity Failure Pattern Observed | |
| --- | --- | --- |
| | Time of Day (Day/Night) | Time of Year (Dry/Wet Season) |
| EHS-Type 1 | ✓+ | ✓ |
| EHS-Type 2 | ✓- | ✓+ |
| EHS-Type 3 | ✓ | ✓- |
| EHS-Type 4 | X | X |

In the hybridized EHS-Type 3 and EHS-Type 4 systems, capacity shortages are generally less and fail randomly, with different probability distributions. Specifically, EHS-Type 3 (solar and wind energy) has a 10% energy capacity shortage, and a time shortage of 6.6%, but failures happen less often during the day. However, EHS-Type 4 (grid and back-up diesel generator) has a 4% time capacity shortage, which was basically distributed during the day and night evenly. The significance of the results are not based on specific energy systems described, but on the methodologies' ability to transcend the specific energy system design. These four EHS types represent four vital electricity failure probability patterned cases facing the global medical community as outlined briefly in Table 1 below.

Although these four regions and EHS-Types may or may not apply to a specific health care facility, they represent random and non-random failure statistical types. EHS-Type 1 has a failure pattern observed during the Time of Day and Year whereas EHS-Type 4 does not have either. One might argue EHS-Type 4 is truly random. However, EHS-Type 2 seems to have a failure pattern change focused on a Time of Day whereas EHS-Type 3's pattern change is between Time of Year. Together, these represent distinct failure patterns of interest to the medical community.

A utility grid with back-up diesel generator is the most common electricity system in healthcare facilities [46, 47], including in the United States. In fact, during the 2003 Northeast blackout [30, 48, 49], Hurricanes Katrina, Sandy, and Maria [41, 50–52], medical disruptions and patient deaths, associated to electric grid failure, were recorded, especially where all the hospital diesel generators failed. One particular US anesthesiologist laments that with all of his medical training, his medical training proved insufficient mid-surgery when he was constantly told to finish up because the batteries were depleting charge and not knowing what to do if they fail. This outlines the importance of intermittency. The aforementioned tragedies underscore the necessity that all prioritized medical devices, that require electricity, have a back-up on-demand energy system [34, 53, 54]. This means that for limited lighting, a gravity-based system could be used, where a medical professional lifts a weight which as it falls over 5-30 minutes, LEDs minimally light the room, improving conditions by eliminating total darkness. This system presupposes that gravity is always available. Thus illuminating the need for a battery and the consequent worry of depleting charge and not being able to see. However, it has obvious power limitations. For slightly more power, a back-up human powered electricity system could supply emergency electricity for anesthesia machines, but not for high powered surgical lighting [26, 55]. Overall in the ladder towards 99.999% reliability, backup and hybridized power systems can be a viable source of energy to combat additional risks associated with capacity shortages.

## 3.2 Risk chart based on hidden VSL/E costs

Although human power has been suggested by a number of researchers, and is implemented in NICUs in emergency situations, there has not been a systematic cost-benefit analysis [53].

Human power, also known as the cost of electricity produced by human powered devices, can affect the VSL/E calculation. Suppose that a typical electricity cost from the centralized grid is $0.10/kWh or that a new designed energy system is calculated to result in a levelized cost of electricity (LCOE) of $0.25/kWh or $0.45/kWh: defined as capital and operational costs over the lifetime of the system. The cost of electricity produced by human powered devices [55] can be significantly less than the VSL/E calculation, either due to acceptable capacity shortage to save costs [31], or neglected power consumption growth [37]. These differences can affect the ability, or lack thereof, to model shortages. The goal of Fig 5 is to illustrate how easy it is to model the risks given, the size of health care facility (number of patients), the percentage of medical procedures dependent on electricity (% of medical procedures with high, medium, low and no impacts), 2 parameters defining the uncertainty of these risks (c and k), and energy system profile types (generated on-site or using sample energy system profile). With this information, a medical professional can change these values and calculate the risk, affording them the information to make medical decisions based on electrical data. Decisions can, and should be, made with models when data does not exist [30]. This means creating a EHS-Risk Chart for ease of understanding the medical consequences to decisions.

Fig 5 shows hidden health care costs, which are 10 to 10,000 times higher than LCOE, that can be modeled to improve medical decisions. These results also apply to a health care facility when it receives medical equipment from donors which increases the power load quickly to a level of 20% energy capacity shortage. Although this analysis includes uncertainty, this chart shows only the results from the mean, for clarity.

## 4. Discussion

Frequent power outages greatly disadvantage resource-poor health care facilities seeking to deliver quality health care [3–15]. There is an unexplored phenomenon whereby health facility executives, due to concerns of fiscal responsibilities and reputation consequences, fail to collect or bypass data on patient outcomes due to electricity failure. As a result, there is a scarcity of data collected in association with electrical failure and patient outcome [29]. However, it is a worthwhile medical issue as it deals with human life and ethical dilemmas. Using two novel methodologies, the authors calculated patient risks when a medical procedure is started and a facility experiences an electricity failure. These methodologies depend on the frequency and duration of an outage as well as medical procedure impact group, type of energy system and location, and the number of patients receiving the medical procedure. Based on the aforementioned information, the number of deaths associated with electrical failures can be quantified and the costs associated with these statistical lives lost based on the region's VSL is calculated and defined as VSL/E. A rather applicable example of the utility of this model is its ability to calculate the feasibility of a claim. A Ugandan newspaper claimed that doctors from Jinja Referral Hospital collected data on 150 patient deaths in six months that were a result of electricity failure alone [56] (equivalent to 300 patients in a year). According to the paper, these patients died due to electricity failure, even with a backup diesel generator. Using this model, the yearly deaths were calculated using the low 4% electricity failure profile from EHS-Type 4. The model was run and quickly it was found that yearly deaths could be 224 deaths with a statistical range between [138, 307] deaths from EHS-Type 4. This ascertains the possibility of deaths with the given information about Jinja Referral Hospital: the hospital had 500 beds, medical procedures would be classified as 45% high, 20% medium, 20% low, 15% no impact medical procedures, c and k uncertainties (described in detail in S1 File). Without this model, there would have been no way to corroborate this claim via calculation and analysis. Not only is this model useful in this sense, it also gives hospitals and healthcare facilities the opportunity

# Risk Chart based on Hidden VSL/E Costs

Health Care Facility (Size=150 beds High=5% Med=10% Low=20% No Impacts=65%) and Energy System (EHS-Type 1 and VSL-Iraq)

| Severity | | Improbable | Remote | Occasional | Probable | Frequent |
|---|---|---|---|---|---|---|
| **Catastrophic**<br>High Chance of 2 Deaths during Day | | No events<br>$0/kWh | No events<br>$0/kWh | 27 Days<br>$24,243/kWh | 14 Days<br>$14,996/kWh | 9 Days<br>$9,378/kWh |
| **Significant**<br>Low Chance of 2 Deaths during Day | | No events<br>$0/kWh | No events<br>$0/kWh | 24 Days<br>$12,323/kWh | 2 Days<br>$957/kWh | No events<br>$0/kWh |
| **Moderate**<br>High Chance of 1 Death during Day | | No events<br>$0/kWh | No events<br>$0/kWh | 19 Days<br>$6,728/kWh | 1 Day<br>$412/kWh | 1 Day<br>$412/kWh |
| **Minor**<br>Low Chance of 1 Death during Day | | No events<br>$0/kWh | No events<br>$0/kWh | 14 Days<br>$4,061/kWh | No events<br>$0/kWh | No events<br>$0/kWh |
| **Negligible**<br>Extremely Low Chance of 1 Death During Day | | 47 Days<br>$739/kWh | 121 Days<br>$12,057/kWh | 65 Days<br>$13,421/kWh | 11 Days<br>$846/kWh | 9 Days<br>$0/kWh |

Legend:
- ▮ (red) Below + considering On-demand Energy Systems
- ▮ (pink) Below + Postpone Procedures
- ▮ (yellow) Below + Check Energy Stored before Starting Procedure
- ▮ (light green) No Change to Procedure*
- ▮ (dark green) No Change to Procedure: Traditional Backup

\* Strongly consider adding another backup system for hybridization and diversification of energy systems before risks increase.

| Improbable | Remote | Occasional | Probable | Frequent |
|---|---|---|---|---|
| 0-2 hrs Time in a Failure Event during Day | 2-5 hrs Time in a Failure Event during Day | 5-10 hrs Time in a Failure Event during Day | 10-15 hrs Time in a Failure Event during Day | 15-24 hrs Time in a Failure Event during Day |

**Likelihood**

**Fig 5. Risk chart showing (a) number of days in a year when health care facilities experience additional risks to patients due to electricity failures (chance of death and likelihood of electricity failure), and (b) VSL/E defined as hidden costs associated with costs of statistical lives lost (VSL: $) divided by energy shortage (E: kWh).** This health care facility has 150 beds; 5% of medical procedures that are highly impacted by electricity failures, 10% medium impact, 20% low impacts, and 65% of medical procedures with no impact due to electricity failure; and uncertainty parameters (c,k) in additional patient risk function explained in S5 Fig in S1 File. This data from the publication in Solar Energy, 2010, sited in rural Iraq was scaled up to a hospital (see S1 File for other ESH types). Note: the LCOE range used by engineers is between $0.05/kWh and $11/kWh and yet these non-zero costs are between $412/kWh and $24,243/kWh.

to adapt, scale, and verify the model and costs to their specific situation to create risk charts that will empower them to evaluate ethical dilemmas due to severity level and likelihood level events. To further illustrate the gravity and pertinence of additional patient risks due to shortages, human rights violations have been brought up against UMEME, the main electricity distribution company in Uganda [57, 58]. Furthermore, doctors were charged with murder after Hurricane Katrina in the U.S. for their role during the outage due to diesel generators placed in flooded basements [52]. It is, therefore, crucial that statistical modeling be available to support doctors' observations and/or ethical dilemma decisions.

## 5. Conclusion and recommendations

The Lancet Commission on Global Surgery, the World Health Organization, the United Nations Sustainable Energy for All, various researchers and organizations have recognized the need for electricity in effective surgical outcomes and in overall global health care. When

developing policies, Ministries of Health and other key players should not only include calls for accessible electricity, but also for reliable electricity. There should be extensive considerations of hidden electricity costs and the ethical dilemmas that medical professionals face when they have to make decisions due to electricity failures. Currently, global electricity poverty has engendered the unimaginable necessity for healthcare professionals to make decisions regarding the continuation, or cessation of procedures, during electricity failures based on anecdotal previous experience. This article makes recommendations to facilitate decision-making by creating an adaptable, scalable, and verifiable model, with accompanying EHS-Risk chart. Medical professionals can be better informed and, therefore, equipped to handle such decisions at the intersection of health care and energy systems.

We propose five global recommendations to achieve the above mentioned goal of informed and ethical decision making:

1. We propose hospitals require every surgical room to post (or surgeon to possess) energy storage levels, obtainable with simple-to-use and easily available sensors. The levels should be presented as either a display in the surgical room or smart phone app. Based on power consumption of their specific medical equipment used for a particular medical procedure (drop down menu from on-line database or preferably local measurements of voltage and current for min, mean, max power needed in kW) and time taken to complete the medical procedure (min, mean, max in hours based on medical team's experience), posted energy requirements for specific medical procedures can be displayed by a smart phone app or calculated and then posted locally (min, mean, max in kWh). If medical professionals knew how much energy was required for a medical procedure (say 1000 kWh), and they knew how much energy was in the battery (say 1100 kWh), in the tanks connected to the diesel generator (say 3000 kWh), or that is about to be stored in the battery from the sun over the next four hours (say 1500 kWh), then they could decide whether or not to even begin said procedure and make backup plans if the procedure takes longer. This is especially true for high impact medical procedures. Furthermore, if they have access to the \$/kWh operational cost as well as the VSL/E cost, then they can consider cost-benefit analysis when choosing between executing or postponing a medical procedure. There is also an opportunity to collect data for R&D funding that would arise from documenting patient outcomes during electricity failures.

2. All hospitals require on-site and/or consultant energy engineers to present the energy system profile for the entire year (only on/off patterns in terms of hour of day and day of the year), especially for single energy source systems like EHS-Type 1 and EHS-Type 2 based on a set of vital uncertainties. Specifically, this requires that they present capacity shortages, in frequency and duration, as it relates to energy system model uncertainties of power load and temperature increases. For example, there are power load uncertainties in terms of new medical equipment, increase in patients and medical procedures. Medical professionals must make their decisions based on frequency and duration of failure events as well as EHS-Risk charts. This allows the medical professionals to calculate the VSL/E and compare this value to the LCOE to make decisions based on a cost-benefit analysis using EHS-Risk Chart (see S1 File for A-E steps). Furthermore, they can require donors of medical equipment to understand the consequences to power load increases based on the EHS-Type with respect to their newly donated equipment. Finally, when energy components are less efficient over time than original design model, health care providers will move from unexpected electricity failure pattern profiles towards documented electricity failure pattern profiles and can make informed decisions in terms of frequency and duration of failures, especially when combined with Recommendation 1 above. [note: energy models neglect

known and documented increases in inefficiencies over time either due to (1) lack of energy modeler's experience for specific parameters—like increases in temperature and power load, (2) lack of quality checks on imported energy systems, (3) lack of consequences to manufacturers or donors, (4) postponed maintenance timetables typically due to financial constraints and no previously available cost-benefit analysis capabilities, or (5) completely neglected serious overtime growth of energy system components' inefficiencies as a result of either extreme temperatures on solar panels installed on metal roofs or without adequate air flow, neglected lack of climate control for room with batteries and charge controller, and/or neglected lack of climate control for room with diesel generator(s) enclosed in secure locations sometimes with no little to no air flow]

3. Overall, in all conditions, we propose further hybridization of all EHS types when risk charts have events in yellow and/or red areas: beginning with on-demand electricity generating systems, for highest priority power loads, where the probability of patient death or injury is high and the amount of electricity needed is low (corresponding to high VSL/E). For minimal security lighting and night births, a gravity powered system can be locally built and maintained (which is safer and cheaper than kerosene lanterns). For some efficient low power oxygen concentrators and anesthesia systems, human powered generators can be implemented, but designed within limits of arm or leg muscles (as dramatized in the film Hours [59]). Health care credit or income could be given to people using an exercise gym, bringing food waste or incineration-capable waste, within limits. Furthermore, waste water treatment options for sanitation to power already existing petrol generators by retrofitting them into biogas generators, concentrating solar power generators or waste incinerators with steam turbines resulting in boiled water, as well as thermal electric generation from cooking are also viable hybridized EHS options that are currently not seriously considered. These energy source options represent a ladder of power level prioritization and opportunities. They could have extensive positive impact depending on EHS-Risk Chart results and VSL/E costs calculated moving forward.

4. Given that electricity failures occur, checklists should be expanded for medical procedures and health care services postponed [60]. Preparations should be made to communicate a proactive plan moving forward for the medical staff, patients and families including surveys for data collection and requests for tracking patient outcomes during electricity failures.

5. Finally, and as a slight side note from the authors, we propose that all outlets should be disabled if not being used for a medical procedure to prevent personnel from increasing power load (consumption). This is a recommendation based on observations made by authors while working or visiting various health care facilities. The observation showed that when doctors and nurses are not adequately paid, some have allowed their health care facility to become a public charging station for cell phones, batteries to be used at homes and businesses, and laptops.

As power loads increase, any energy healthcare system will cross a threshold and go from 0% capacity shortage to 20% capacity shortage—sometimes quickly. This shortage can happen randomly during the day when the power load varies the most. Specifically for solar panel systems, this is contrary to thinking the shortages would happen at night. The shortages will happen when the power load grows. So, a 20% yearly capacity shortage could medically feel like a 40% capacity shortage during the day (or almost a coin toss level of random electricity failure events, see EHS-Type 1 result). Connections between electricity failures and health care consequences can now result in generating EHS-Risk Charts.

## 6. Future EHS research and direction

Future EHS research must focus on data collection methodologies that allow healthcare professionals to quantify additional risk to patients as well as the creation of a database on electricity failures. This database will be used to generate more EHS-Risk Charts which together will provide maps for decision-making. Suppose one has a fuel tank/electrical energy gauge for medical health care facilities and/or surgical rooms. Then think about going to a database, via an app, where one selects the devices and how long they will be in use, allowing the database to map out to the medical professional the equivalent energy system use. This is similar to going to Google maps for directions and travel distance, but in terms of time of procedure and medical devices used. In other words, this model would give the medical professionals the approximate time within which to finish a procedure, the total energy use required for the medical procedure, and any uncertainty involved. This will equip the professional with information to decide whether to begin a medical procedure or to postpone it based on a EHS map. A person can plan a trip in a car where there are few filling stations available. There are many variables that contribute to how fast a vehicle's tank is depleting, including speed of vehicle, and acceleration/deceleration events. Overall, drivers around the world use this type of mapping technology to determine the parameters of their trip. Just as in traveling long distances, there are many times when walking or flying in a plane makes absolutely no sense, but an interesting option to consider. Creating EHS maps based on various EHS-Risk Charts can guide medical professionals in mapping out medical procedures according to energy system options, constraints, and consequences. However, currently the medical community does not posses such a map to provide guidance during an electricity failure for a given medical procedure. The EHS-Risk Chart comparisons hold within them the pathway possibilities and VSL/E costs allows a comparison to LCOE in making ethical medical decisions in energy system options and choices.

## Supporting information

**S1 File.**
(PDF)

**S1 Data.**
(XLSX)

**S2 Data.**
(XLSX)

**S3 Data.**
(XLSX)

**S4 Data.**
(IPYNB)

**S1 Fig.**
(TIFF)

**S2 Fig.**
(TIFF)

**S3 Fig.**
(TIFF)

**S4 Fig.**
(TIFF)

**S5 Fig.**
(TIFF)

**S6 Fig.**
(TIFF)

**S7 Fig.**
(TIFF)

**S8 Fig.**
(TIFF)

**S9 Fig.**
(TIFF)

**S10 Fig.**
(TIFF)

**S11 Fig.**
(TIFF)

**S12 Fig.**
(TIFF)

**S13 Fig.**
(TIFF)

**S14 Fig.**
(TIFF)

## Author Contributions

**Conceptualization:** Abigail Mechtenberg, Brady McLaughlin, Michael DiGaetano, Moses Musaazi.

**Data curation:** Abigail Mechtenberg, Brady McLaughlin, Emmanuel Etwalu.

**Formal analysis:** Abigail Mechtenberg, Lydia Nanjula, Mark Shrime.

**Funding acquisition:** Moses Musaazi.

**Investigation:** Abigail Mechtenberg, Michael DiGaetano, Abigail Awodele, Leslie Omeeboh, Emmanuel Etwalu, Lydia Nanjula, Moses Musaazi.

**Methodology:** Abigail Mechtenberg, Brady McLaughlin, Michael DiGaetano, Moses Musaazi, Mark Shrime.

**Project administration:** Abigail Mechtenberg.

**Resources:** Abigail Mechtenberg, Leslie Omeeboh.

**Software:** Abigail Mechtenberg.

**Supervision:** Abigail Mechtenberg.

**Validation:** Abigail Mechtenberg, Emmanuel Etwalu, Lydia Nanjula, Mark Shrime.

**Visualization:** Abigail Mechtenberg, Abigail Awodele, Lydia Nanjula, Mark Shrime.

**Writing – original draft:** Abigail Mechtenberg, Brady McLaughlin, Michael DiGaetano.

**Writing – review & editing:** Abigail Mechtenberg, Abigail Awodele, Leslie Omeeboh, Emmanuel Etwalu.

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
