## [Decision Letter · Decision Letter 0]

8 Apr 2020

PONE-D-19-36008

Impact of electricity failures on health care in terms of risks from hidden costs: A value of statistical life lost divided by energy shortage model events into VSL/E

PLOS ONE

Dear Dr. Mechtenberg,

Thank you for submitting your manuscript to PLOS ONE. After careful consideration, we feel that it has merit but does not fully meet PLOS ONE’s publication criteria as it currently stands. Therefore, we invite you to submit a revised version of the manuscript that addresses the points raised during the review process.

We would appreciate receiving your revised manuscript by May 15 2020 11:59PM. To enhance the reproducibility of your results, we recommend that if applicable you deposit your laboratory protocols in protocols.io, where a protocol can be assigned its own identifier (DOI) such that it can be cited independently in the future. For instructions see: http://journals.plos.org/plosone/s/submission-guidelines#loc-laboratory-protocols

We look forward to receiving your revised manuscript.

Kind regards,

Fausto Cavallaro, PhD

Academic Editor

PLOS ONE

Additional Editor Comments (if provided):

Dear Dr. Abigail Mechtenberg,

As suggested by the reviewer the paper needs a further minor revision.

Journal Requirements:

"No."

"No authors funded to do this specific research project due to the fact that funding

agencies we approached do not fund this type of research connecting electricity and

health care directly."

Reviewers' comments:

Reviewer's Responses to Questions

**Comments to the Author**

1. Is the manuscript technically sound, and do the data support the conclusions?

Reviewer #1: Yes

Reviewer #2: Yes

2. Has the statistical analysis been performed appropriately and rigorously? 

Reviewer #1: Yes

Reviewer #2: Yes

3. Have the authors made all data underlying the findings in their manuscript fully available?

Reviewer #1: Yes

Reviewer #2: Yes

4. Is the manuscript presented in an intelligible fashion and written in standard English?

Reviewer #1: Yes

Reviewer #2: Yes

5. Review Comments to the Author

Reviewer #1: 1. EHS is first mentioned in the introduction in line 25, but is not explained until line 33.

2. “Three of these were published as either optimal 46 or politically necessary for an energy healthcare system (EHS-1,2,3 type) solution, and 47 discussed with 10-50% capacity shortage (failure events).” This sentence is confusing and needs to be explained more fully. Where is the data coming from and why is the data from Uganda obtained using a different methodology.

3. The authors present a novel method for providers to evaluate risk for surgical patients in health facilities

with unreliable electricity. This is a valuable tool and useful addition to the literature.

Reviewer #2: In this paper, author(s) developed two methodologies to quantify these hidden costs. I think the paper is well written and its structure is well organized, and the study is innovative. So, I think this paper can be published in the current form.

6. PLOS authors have the option to publish the peer review history of their article (what does this mean?). If published, this will include your full peer review and any attached files.

Reviewer #1: Yes: Sagar Chawla

Reviewer #2: No

---

## [Author Response · Author response to Decision Letter 0]

8 Jun 2020

These are also in the response letter.

Reviewer #1: Our response

1. This has been addressed by the author(s) who provided an explanation of EHS on line 25 and omitted the explanation on 33. 

2. This sentence had been corrected as follows :

Three of the papers, cited above, published solutions for EHS 1,2,3. These solutions are presented as optimal, or necessary, given the political climate of the location. Furthermore, they presented and accepted a capacity shortage of 10-50\\%. These solutions were replicated and analyzed to delineate why such huge margins in capacity shortage were accepted, given the consequences to patients. The fourth system, EHS 4, in Uganda is based on actual measurements of voltage and current over half a year. The data from Uganda was used to create a contrast to modeled data. This is to say that it adds to the flexibility and applicability of the paper as it is not limited to modeled data. In this way, all the data were used to quantify and better understand the factors that contribute to accepting capacity shortage in an EHS.

3. We thank the reviewer for this comment. The paper has undergone a rigorous editing phase to allow for this level of clarity from a reviewer. 

Reviewer # 2: Our response

We thank the reviewer for the comment. We have made the revisions suggested to allow for publishing. We hope this newest copy will be published in its current form.

---

## [Editor Report · Decision Letter 1]

23 Jun 2020

Health Care during Electricity Failure:  The Hidden Costs

PONE-D-19-36008R1

Dear Dr. Mechtenberg,

We’re pleased to inform you that your manuscript has been judged scientifically suitable for publication and will be formally accepted for publication once it meets all outstanding technical requirements.

Kind regards,

Fausto Cavallaro, PhD

Academic Editor

PLOS ONE

---

## [Editor Report · Acceptance letter]

26 Jun 2020

PONE-D-19-36008R1 

Health Care during Electricity Failure:  The Hidden Costs 

Dear Dr. Mechtenberg:

I'm pleased to inform you that your manuscript has been deemed suitable for publication in PLOS ONE. Congratulations! Your manuscript is now with our production department. 

Kind regards, 

on behalf of

Professor Fausto Cavallaro 

Academic Editor

PLOS ONE